# Analysis of Factors Affecting the Circularity of Building Materials

**DOI:** 10.3390/ma14237296

**Published:** 2021-11-29

**Authors:** Joanna Sagan, Anna Sobotka

**Affiliations:** Faculty of Civil Engineering and Resource Management, AGH University of Science and Technology, 30-059 Kraków, Poland; sobotka@agh.edu.pl

**Keywords:** sustainable development, circular economy, construction waste, circular construction

## Abstract

A circular economy requires closed circuits of consumed resources. Construction generates approximately 50% of solid waste globally, which is difficult to manage. The aim of this article was to identify the factors that determine the development of circular construction in the context of waste minimisation in the life cycle of building structures. The identification of cause-and-effect relationships by means of the DEMATEL method allows the problems of construction waste management to be taken into account in the context of the development of sustainable construction and fulfilling the principles of the circular economy.

## 1. Introduction

The development of human civilisation observed in recent decades based on a consumer lifestyle ensured by the production of low-quality products without respect for environment has led to a perceptible degradation of the natural environment [1,2]. The result of this situation is a complete change in the concept of social and economic development, aimed at finding a balance in the economic, social and environmental fields. The specific targets set by the UN for 2030 [3,4] are intended to help achieve this transformation parallel in all countries [5]. For the chosen topic of consideration, Sustainable Development Goal 12 takes on special significance [6]. Responsible consumption and the dissemination of production patterns based on agile supply chains put the circular economy into practice. As a result, waste minimisation and reducing the consumption of natural resources becomes conceivable [7].

It can be seen that traditional construction has, for centuries, used durable building products, buildings and structures have been constructed to last for centuries and materials from demolition such as bricks were reused. In the 20th century, this circular economy-like model moved away towards a linear economy. For example, by using technologies for building structures (houses) that are less durable, difficult to dismantle, without well thought-out solutions for waste management or adaptation of the facility to new utility functions. The main objective was to build as many residential and industrial buildings as possible in the shortest time. Currently, the concept of circular economy is a panacea for the problems of the depletion of natural resources and increasing amounts of waste.

The article presents conditions for the implementation of the circular economy in the construction industry (experience gained in Poland), with a focus on waste minimisation and re-use as the basis for circular economy in the construction industry. To this end, the drivers and constraints of circular construction were defined and, using the decision making trial and evaluation laboratory (DEMATEL) method [8], causal relationships were identified for consideration in the management of construction projects. The results can be used to allocate financial resources (e.g., through EU-funded project support) to stimulate the market to move towards circularity. 

## 2. Determinants of the Construction Materials Circularity and Waste Minimisation

### 2.1. Reverse Logistics

The core tasks of implementing circular economy in construction include reverse logistics (RL). This covers the last (closing) links in the closed supply chains of the circular economy. Nobody is mentioned as the creator of the RL concept, although many of its assumptions and theoretical underpinnings were derived from reports developed by Stahel and Reday [9,10]. A closed loop supply chain has been defined by [11] as the process of designing, controlling and operationalizing a system for maximizing the value created throughout the product’s life cycle with the ability to dynamically recover value from different types and quantities of returns.

The closure of the classical supply chain can take place at different levels, extracting differentiated product value [12]. RL is considered an inseparable component of closed-loop supply chain (CLSC) [13], which is also key to building a highly efficient supply chain [14]. 

In the context of the benefits of re-engineering supply chains within the manufacturing industry, supply chains in construction should also be reorganised to a closed form [15,16,17]. Materials and construction products from an out-of-use building should be recovered as whole elements, parts or materials to be incorporated into the structure of a new building [17]. The proper flow of materials in construction should reflect the natural mechanisms of resource efficiency without generating waste [18,19]. 

Basically, reverse logistics was concerned with the return of goods between consumer and producer, but a reverse logistics system can take a much more complex form. Bai and Sarkis [14] point out that these reverse flows do not necessarily mirror the classic supply chain, although they often (in manufacturing industry) take this form. In real systems, materials and products can be directed via diverse channels while remaining within the reverse logistics system [20]. Brito and Dekker [21] and Mafakheri and Nasiri [22] emphasise that the reverse logistics system does not have to close at the point of origin of the product, but at any place where its value can be recovered. 

The primary methods for recovering raw materials and construction products are: reuse, repair/renewal, parts extraction/recovery, and recycling—turning waste into a raw material. Additionally, in the design stages, the technical and material characteristics of the building can be shaped, adapting it for easy disassembly and the recovery of materials from demolition [23].

However, despite the experience of the manufacturing industry, achieving the benefits of closing supply chains in construction [24,25] is fraught with much greater difficulties. Among barriers, the interdisciplinary nature of recovery activities, wide range of variables conditioning the flow of return streams [26], and limited knowledge and practice in the field are highlighted [27,28]. The literature points out the lack of a broad industry identity with commensurate standardized practices [15], and indicates that the provision of information can act as a driver of recovery growth [16].

Also noted was the particularly long lifespan of a building object in contrast to those of the products of the factory industry.

The main (internal) barriers to the implementation of the RL in practice, include insufficient knowledge regarding the organisation of reverse supply chains [29]; uncertainty of the results of the implementation of RL [27,30]; resistance of organisations to change developed patterns of behaviour [31]; scarcity of resources including financial to cover initiation costs [27], including infrastructure and manpower requirements of the reverse logistics [32]; and poor product quality [33]. 

Some barriers also concern the surroundings of an organisation. These include: customer reluctance towards recycled products due to insufficient knowledge of product quality [30]; lack of government support [29]; and lack of product adaptation for recovery [34].

Among the specific barriers occurring in the construction industry there are: problems of organising return chains due to limited construction site space with particular time-consuming RL processes [28]; lack of knowledge about the benefits of RL implementation [15]; large size of waste [18]; long product life cycle [33]; unsuitability of existing facilities for deconstruction [16]; variable waste characteristics [35].

Identification of barriers and RL practices to construction and demolition waste can lead to plans to mitigate them [36].

### 2.2. Sustainable Design

Until recently, functional and structural issues were the dominant aspect in the design of buildings. The priority was to obtain a structural solution with the required characteristics, guaranteeing that the needs of the investor and users of the building would be met [37], while keeping the investment costs as low as possible. The time frame, within which the effectiveness of the adopted solutions was considered, only included the stage of construction of the building and the guarantee period. The new approach in design, incorporates all the requirements for sustainable construction into a single design process, called the integrated life cycle design (ILCD) [38,39] (Figure 1). ILCD takes a holistic view of the entire life cycle of a facility, also with regard to construction waste management, taking into account the waste hierarchy in the process of future waste management.

The new approach takes into account [40,41]:Various aspects of design (functional–structural, sustainability, environmental, economic, social and cultural);Subsequent phases of use throughout the design life, i.e., starting with the procurement and production of materials or components, through construction, use and maintenance, and ending at demolition, recycling and waste disposal;Different levels of optimisation (materials, components, elements, whole structures) taking into account a set of selected criteria including those from the aforementioned groups, i.e., environmental, economic, technical, and social and cultural.

When considering an object in its life cycle and, in particular, taking into account its demolition and recycling and, in fact, waste disposal (reuse of elements, raw materials and recycled materials), the sustainability (including quality) of elements and whole objects takes on a wider meaning (Figure 2). 

Durability is the period during which a structure (product, component, building object) retains its performance characteristics. Structural durability, alongside safety and serviceability, was analysed as one of the three pillars of structural reliability. The aspect of durability in the design of, e.g., concrete structures, recently more often referred to as design for life (although not always equated), is based on physical and chemical tests, where chemical issues are predominant [42]. 

### 2.3. Assessment of the Condition of Building Elements and Structures 

Structures and their components (structural systems, embedded building materials, façade materials, installations, etc.) are exposed to many destructive factors, such as changes in temperature, humidity, static and dynamic loads or chemical and biological aggressions. Therefore, the quality of the product from recovery for reuse or recycling for secondary raw materials may be different from the initial product quality—hence it needs to be assessed [43,44,45]. 

The determination of the physical and mechanical properties of materials can be effectively carried out using modern testing methods such as non-destructive methods. Within non-destructive testing, the following methods can be distinguished: organoleptic testing (macroscopic evaluation) and any technical testing based on acoustic, electrical, radiological or electromagnetic methods [46,47]. Within the scope of non-destructive and semi-destructive tests, there are also tests based on different types of mechanical impact such as the measurement of the pull-out force and the depth of the induced defect [48,49,50]. The basic features of building elements to be assessed are: the shape and dimensions of the element, the material properties, the location and extent of damage, and the corrosion of the material including microbiological assessment [51].

Failure to adequately test an element before its re-use can lead to a construction failure. Such situations exacerbate barriers to the use of recycled products, but on the other hand, the use of non-destructive testing requires specialist knowledge and practical experience [44]. 

### 2.4. Provisions of the Law

A waste producer is whomever that generates waste as a result of their activity or livelihood. A waste generator is therefore an entity that provides services as part of construction work (construction, renovation, demolition), except where the contract for the provision of services provides otherwise.

Waste must be managed in such a way as to protect human health and life and the environment in accordance with the waste hierarchy [52].

The prevention of waste generation also includes activities related to the design of renovation, taking into account the strengthening rather than the replacement of existing elements, the renewal of window frames, roof tiles cleaning, etc. These processes are especially practised in the renovation processes of historical buildings. 

In cases when the generation of waste cannot be prevented, the producer of that waste shall carry out recovery, i.e., preparation for reuse or recycling. However, in order to be able to recover waste by processing it, the safety of the intended activities for the environment must be guaranteed, and an appropriate waste processing permit must therefore be obtained from the competent authority, e.g., the Marshal of a Province, a Starost or the Regional Director for Environmental Protection, depending on where the waste is managed. However, obtaining a permit that covers the implementation of waste management processes on specific building plots requires an administrative procedure that, in practice, takes between 2 and 4 months (from the date of application to obtaining the permit). The time taken for the preparation of the relevant documents must be added to this time. Such documents include the application itself, the fire safety report, the declaration of no criminal record, and may include the water permit and the decision on environmental conditions. 

There is currently an ambiguous interpretation of the law in practice. From the perspective of construction work contractors, as long as they do not want to get rid of a given raw material from the construction site, they do not classify it as waste and therefore do not subject it to processing but they process only the building material, e.g., to reduce the granulation of rubble to increase the efficiency of transport processes. In many cases, the controlling authorities take a different view, for whom excavation spoils and demolition materials are waste and the crusher is a processing plant. The problem is exacerbated when the waste storage and treatment site are located on neighbouring (leased) land due to insufficient construction site space. In many cases, these parcels are not covered by a permit or a decision on development conditions. It is therefore necessary to transport the waste to the sites covered by the permit. This, in turn, contradicts the proximity principle, which states that waste should first be treated at its place of origin.

When reading the law straightforwardly, one may arrive at the conclusion that processes for cleaning and repairing construction waste (wood treatment, sandblasting or cutting brick waste, etc.) are also waste treatment processes which thus should be subject to authorisation. The need to obtain the relevant permits prolongs and hinders the development process, while the process of demolishing a building can take several days. 

### 2.5. Lean Management

One of the ways of contributing to waste minimisation is the production and management method known as lean management (LM) [42,53]. One of the aims of LM is eliminating waste in every area of the business. Lean management uses various modern management concepts aiming, in general terms, to use the environment in an economical manner, closed resource cycles, reducing human effort and non-value-added activities [54,55,56].

Lean production can mean the economical production of a previously designed product. At that time, the basic principles applied in the early stages of implementing this method were: on-time delivery; minimisation of stocks; utilisation of the company’s production capacity; shortening of production cycles. 

The concept of lean production in construction should be applied to all phases of the product life cycle of a building, i.e., the production of materials and products for construction, design of facilities, execution and operation, and finally the decommissioning process. In the latter stage, it is a question of ease of reconstruction, ease of demolition, reuse in another building, ease of separating the individual elements of the building and the layers that make up the building element—the segregation of materials. Finally, recycled materials such as raw materials can be used for the manufacture of new products for the construction industry or other sectors of economy. Material engineering plays a major role in this respect with regard to providing new materials using recycled materials and convincing participants in construction projects to use them [57,58].

The concept of lean construction should take into account the principles of sustainable development in the sense of saving natural resources, using energy from non-renewable sources and not emitting harmful compounds during the life cycle (during manufacture, construction and operation) of facilities. The implementation of these principles—lean production—requires the application of the well-known and developed lean management method in economic organisations (Figure 3).

### 2.6. Others

A product such as a building, which arises as a result of a construction project, is created and used in the different phases of its life cycle by the changing participants in the project. The problem is the integration of their objectives and the coordination of their activities in the “development and delivery of the product”. According to the concept presented by the article authors, the strategic aim of these activities should also be to care for the environment by minimising waste [57,59]. Another way to reduce waste is to develop new products that are manufactured and operated with waste reduction in mind. This requires additional investment in waste reduction resources (raw materials, production technologies, equipment, personnel, research—etc.) [60]. A supporting method for these activities is the so-called supply chain management (SCM) [61], in which there is also an opportunity to implement lean management principles [62]. In contrast, institutional pressures and state policies are necessary for full success in achieving this goal. 

## 3. Cause-and-Effect Analysis of the Construction Material Circularity and Waste Minimisation

### 3.1. Methods

Taking into account the conditions discussed above and taking into account the specifics of construction production in the past and at present, as well as development trends and the analysis of the research literature, 33 factors determining the development of circular construction were listed. 

To investigate the identification of causal relationships in the issue at hand, the authors proposed the use of the DEMATEL method [8]. This method is widely used in the analysis of economic processes in the environmental aspects [63,64,65]. The DEMATEL method identifies the form of the influence structure by means of a cause–effect diagram. Its graphical interpretation provides a clear map of the influence relations of the factors under study. In addition, the total flow arcs (T) are accompanied by numerical values indicating the intensity of the total net effect. Graphical representation and subsequent transformations and calculations using the DEMATEL method make it possible to determine the values of indicators that describe the role and significance of the factors under consideration in the context of direct, indirect and total impact. 

The following calculation procedure was used [8]:To assess the relationships between factors, a direct influence graph was created which defined the occurrence of the influence and its direction;The strength of the impact was presented by the direct influence matrix (Equation (1)) (Appendix A, Table A1). The following scale of impact was used in the analysis: 0—no influence; 1—small influence; 2—medium influence and 3—significant influence:
(1)$$Z={\left[z_{ij}\right]}_{n\times n}$$In the further step, the determination of the normalised direct influence matrix (Equations (2) and (3)) and then the determination of the total influence matrix (Equation (4)) were calculated:
(2)$$X=\frac{Z}{s}\;$$
(3)$$s=\max\left(\underset{1\leq i\leq n}{\max}\overset{n}{\underset{j=1}{\sum }}z_{ij},\underset{1\leq i\leq n}{\max}\overset{n}{\underset{i=1}{\sum }}z_{ij}\right).$$
(4)$$T=X{\left(I-X\right)}^{-1}\;,$$On the basis of the above matrices, the vectors R and C (Equation (5)) were computed, as the sum of the rows and the sum of the columns from the total-influence matrix (Appendix A, Table A2):
(5)$$\begin{array}{c} R={\left[r_{i}\right]}_{n\times 1}={\left[\sum_{j=1}^{n}t_{ij}\right]}_{n\times 1},\\ C={\left[c_{j}\right]}_{1\times n}={\left[\sum_{i=1}^{n}t_{ij}\right]}_{1\times n}^{T}{}_{\;}, \end{array}$$

Above vectors are the element of “prominence” (R+C), and “relation” (C−R), which successively illustrate the strength of the influences of the factor and effect that the factor contributes to the system [8].

### 3.2. Research Findings and Their Analysis

In the first stage of the conducted research, the links between the various factors leading to construction waste management and the main objective of waste minimisation were identified. The structure of the factors and the relationships between them are shown in Figure 4. The arrow represents the cause–effect direction (arrowhead). The type of arrows and their thickness do not describe the relations, as the distinction is only used for clarity of drawing. The relationships between the factors, as well as the magnitude of the relationship can be found in the direct influence matrix (Appendix A).

Then, by applying the assessment of a five-person team of experts (demolition specialists, reverse logistics, construction managers, and academics), a direct influence matrix was developed (Appendix A). The experts were selected because of their competence, both in the field of scientific knowledge and practical experience, which allowed to broaden the scale of observation. All experts were from one country (Poland). The DEMATEL analysis carried out made it possible to draw up a graph presenting the influential relation map (IRM) (Figure 5).

The results obtained present structured factors either directly or indirectly influencing waste minimisation. The information obtained made it possible to identify the relationships between them as cause and effect and determine the strength of these links. The results of the analysis set out those factors to which participants in construction projects, and more broadly in the construction industry (and those associated with it—working for the benefit of construction), must pay particular attention. 

The data show (Appendix A) that the strongest links (and the highest number of links) are between the factor knowledge and practice of waste management as a cause of waste segregation, further quality of secondary raw material, which results in material recovery. Similarly, the use of a sustainable design approach (cause) provides the following effects: the ease of adaptation of objects, the durability of the object (made of durable products and components), and the use of complex materials with components easy to separate and recover.

According to [8], based on the mean value of the prominence, the IRM diagram (Figure 5) was divided into four areas (marked in green). Section I includes those factors that were identified as key factors in the transformation of the construction industry into a closed loop system. This group included socio-economic trends that influence the investor requirements and societal beliefs about recovery which were also in the key factors. 

Section II contains factors identified as additional drivers for the model. This group includes factors related to the integrated design (as lower-level components): interdisciplinary project teams, building certification, environmental product declaration—all easily adopted to changing usage requirements. In this group, there are also many elements of market background, such as investment demand, financial and non-financial sub-measures, environmental protection requirements (legal requirements), complex investment paths for waste management facilities, and investment capital.

The factors in group III are the so-called autonomous receivers which have a low rank and relationship, they are relatively detached from the system. This group included factors such as using BIM technology, lean construction and minimising construction activities during the operational phase to extend the life cycle of buildings.

Section IV includes factors that are influenced by other factors and are more resistant to direct correction, which include, among others: building deconstruction; demand for recycled materials and products; waste management cost; and minimisation of construction waste.

## 4. Summary and Conclusions

The analysis of the results shows that the key element of a circular economy is waste recovery, which is most strongly governed by financial aspects, i.e., the relation between the costs of waste management and the prices of recovered raw materials. Improving this relationship will result in a growing market for recyclers, an observation which is consistent with those of [66]. However, these factors are resistant to direct correction, so their improvement can be achieved by improving key and stimulant factors. The market for recycled raw material is heavily dependent on social beliefs which are driven by social and economic trends and require knowledge and practice in proper waste management. An increase in the mentioned awareness and a guarantee of the quality of the raw material will lead to a change in the requirements of the client and thus to an increase in demand for recycled materials. Sustainable designs also significantly contribute to the development of circular economy. The use of materials and technical solutions that are easy to dismantle enable the selective collection of waste during the deconstruction of the building and the minimisation of waste. 

The recovery of waste is limited by the time and organisational conditions of the site, including the legal background. Financial benefit, strongly emphasised, must be significant to compensate for the additional difficulties in the investment process. To ensure the quality of the raw material, it is important to develop recovery techniques and technologies. Additionally, at the same time, these waste recovery activities must be backed up by the pressure of international and national legislation (institutional pressures).

The lesser importance of such conditions as the implementation of lean construction, life cycle extension, the use of building structures and materials with durable and easily separable and recyclable components is due to the fact that, despite the high quality of waste, the recovery of materials will not be possible if there are no recyclers, no effective waste treatment technologies, or no market for recycled products.

Nowadays, due to the enormous degradation of the environment, methods are being sought for an efficient, “fast”, and long-term sustainable economic model. This is why methods are being developed to support the factors that influence the return to economical construction, the construction of sustainable and self-sufficient buildings and at the same time the use of modern (including waste-free) materials and structures. Researchers and manufacturers are being encouraged to seek and invest in resources to reduce waste and all waste, from products, human labour, equipment, energy and water (further details can be found in [60]). 

To summarise, the following basic directions for the development of circular economy in construction can be identified:–The development of a market for recycled materials and products based on selective waste collection and modern waste processing, resulting in a market with high-quality products that customers will have confidence in;–The implementation of an integrated life cycle design (ILCD) approach with an analysis of the adaptability of buildings (waste prevention), including the recovery potential of the materials used in the design process (easy for recovery); These directions are consistent with the observations presented in [57].

The complexity of the problem was recognised during the study; though only five experts of homogeneous nationality were interviewed in this study. The assessment concerns the current state, and the observations are not timeless. The sensitivity analysis of the model to factor weights was not provided, though this is nevertheless considered a direction for further work, particularly in the field of nationality context.

## Figures and Tables

**Figure 1 materials-14-07296-f001:**
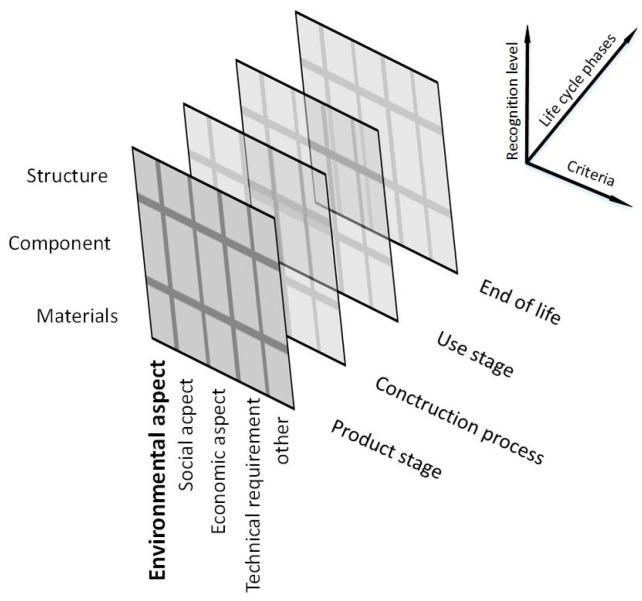
3D model of an ILCD [38].

**Figure 2 materials-14-07296-f002:**
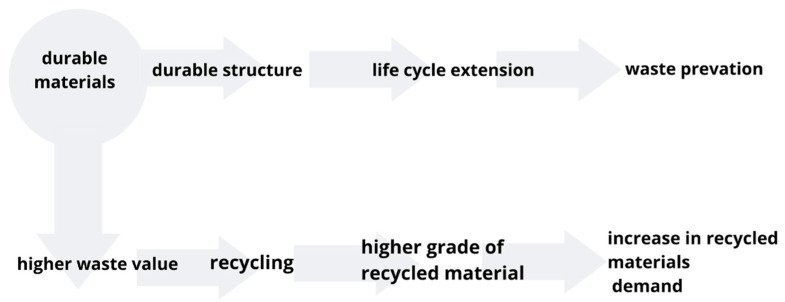
Durability in circular economy.

**Figure 3 materials-14-07296-f003:**
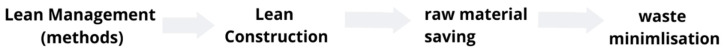
Waste minimisation from lean production concept.

**Figure 4 materials-14-07296-f004:**
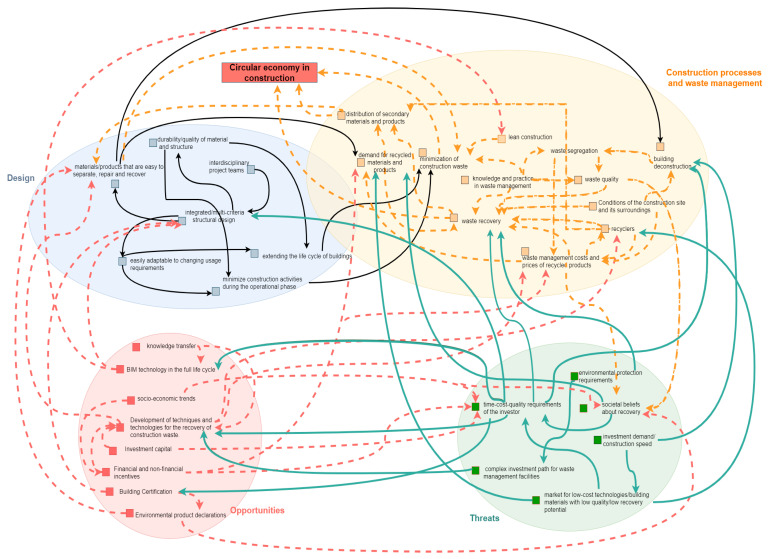
Relationships between factors.

**Figure 5 materials-14-07296-f005:**
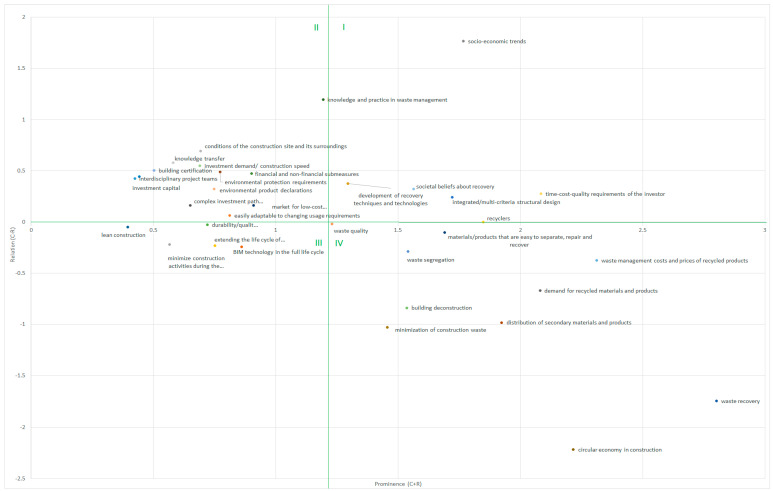
Relationship factor and prominence diagram.

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
