# Peer review of "Analysis of Factors Affecting the Circularity of Building Materials"

_materials, 2021, doi:10.3390/ma14237296_

Round 1
Reviewer 1 Report
It is interesting to receive a contribution on the Title- Analysis of the factors of construction materials circularity and waste minimization.
- Title: It clearly describes the article. However, I feel it is long. It would be better if it could be shortened.
- Abstract: It reflects the content of the article.
- Introduction: The introduction is well formulated, having the background, state of the art review, analysis, and formulation of the goals.
- Provide the full form in the first place, and then it can be followed by acronyms. (Ex. CLSC; ILCD)
- There are reference lumps in the text, such as [13,14,25,26]. Would you please eliminate this lump? After that, please check the manuscript thoroughly and eliminate all the lumps. This should be done by characterizing each reference individually. This can be done by mentioning 1 or 2 phrases per reference to show how it is different from the others and why it deserves mentioning.
- There are only 4-5 recent references. Kindly use relevant recent references from 2019 - 2021. For the credibility of the review, at least 9-10 recent references are needed.
- The direction of arrows in figure 3 is a bit confusing. Please provide explanatory text before the figure to improve the clarity.
- It is highly difficult to read Figure 4.
- The conclusion section needs to be revised and mention the claims from the results.
Author Response
Thank You for your valuable comments that contributed to improving our publication. The corrections and commentary are provided in the word file. The article has been revised in change tracking mode, for easier verification of the changes made.

Reviewer 2 Report
Very detailed and complete review of the literature by the authors. A number of interesting and timely concepts are being explored. It is a very complex problem, so any advance should be commended!
I would ask the authors to clarify the model, providing more detail about the influence factors. Absolutely critical is that you provide more details and context on the experts consulted to generate your results.
Detailed comments:
Lines 261 – 283:
What is the “normalized direct influence matrix” a measure of? I can see mathematically what the authors are doing, but I have no idea what these manipulations mean in a physical context. What does the normalizing process do? More detail needed by the authors.
The authors claim that:
“Above vectors are the element of "Prominence" (R+C), and the “Relation” (C-R), 281 which illustrate successively, the strength of influences of the factor and effect that the factor contributes to the system.” How do you know this? I expected some sort of minimization or maximization process to determine the important influences. How does the method proposed by the authors do this?
Lines 291 – 292:
“Then, by applying the assessment of a five-person team of experts (demolition specialists, reverse logistics, construction managers and academics) a direct influance matrix 292 was developed (Appendix 1).” The authors need to provide much more detail on how this was done. Given the wide scope of the problem, “five” experts does not seem like a lot! Surely there is disagreement on the influence of all of these factors. How were the experts selected? Did they meet as a group and come to consensus, or did you poll them individually? Were all the experts from one country, which would tend to skew their experiences?
How sensitive is the model to changes in the weighting? The authors need to address these questions.
Lines 302 – 304:
“The data shows that the strongest links (and the highest number of links) are between the factor knowledge and practice of waste management as a cause, and separate collection of waste and quality of secondary raw material, which results in more efficient material recovery.” How do you get this from Figure 5? You need to explain the figure in more detail to the reader.
Lines 315 – 321:
“The results of the analysis show a large number of factors of lesser importance for the development of circular economy, these include lean construction, extension of the life cycle of a building, easy adaptation. It is easy to see that the current focus of circular economy is put on waste management rather than prevention. This is due to the current problem of managing already produced materials and constructed facilities, the perspective of long-term prevention - it is important but secondary for the moment. The results obtained are consisted with studies by other researchers e.g. described in [53].”
The authors need to spend some time here talking about the limitations of this work!! Yes, you have identified some likely influential parameters, however, it is really just based on the opinion of five experts! You need to provide more context for how sensitive the analysis is to changes in the influence matrix, and whether expert opinion is needed to make definitive conclusions.
Author Response
Thank You for your valuable comments that contributed to improving our publication. The corrections and commentary are provided in the attached file. The article has been revised in change tracking mode, for easier verification of the changes made.

Round 2
Reviewer 2 Report
I have reviewed the author's revisions, and they address my comments. Therefore, I recommend publication of the revised paper.